# Comparison of a Commercial Enzyme-Linked Immunosorbent Assay (ELISA) with the Modified Agglutination Test (MAT) for the Detection of Antibodies against *Toxoplasma gondii* in a Cohort of Hunting Dogs

**DOI:** 10.3390/ani12202813

**Published:** 2022-10-18

**Authors:** Aicha Yasmine Bellatreche, Riad Bouzid, Amandine Blaizot, Dominique Aubert, Radu Blaga, Khatima Ait-Oudhia, Delphine Le Roux

**Affiliations:** 1Ecole Nationale Supérieure Vétérinaire, Rue Issad Abbes, Oued Smar, El Harrach, Algiers 16000, Algeria; 2Anses, INRAE, Ecole Nationale Vétérinaire d’Alfort, Laboratoire de Santé Animale, BIPAR, F-94700 Maisons-Alfort, France; 3Faculté de Médecine Vétérinaire, Université Chadli Bendjedid El Tarf, BP 76, El Tarf 36000, Algeria; 4National Reference Laboratory for Animal Toxoplasmosis, Laboratoire de Santé Animale, ANSES, F-94700 Maisons-Alfort, France; 5EA 7510, UFR Médecine, University of Reims Champagne Ardenne, F-51095 Reims, France; 6Laboratory of Parasitology, National Reference Centre on Toxoplasmosis, Centre Hospitalier Universitaire de Reims, F-51095 Reims, France; 7Laboratory of Biotechnology in Animal Reproduction, Université de Blida, Blida 09000, Algeria

**Keywords:** *Toxoplasma gondii*, seroprevalence, dogs, antibodies, enzyme-linked immunosorbent assay (ELISA), modified agglutination test (MAT)

## Abstract

**Simple Summary:**

Toxoplasmosis is a zoonotic disease that has serious consequences for immunocompromised individuals, in particular for the fetus during primary infection in pregnant women. The parasite responsible, *Toxoplasma gondii*, can infect all warm-blooded animals, and studies on antibodies against the parasite in their blood can give a good estimate of its burden in the environment and the risk for human infections. This is less studied in dogs than in other animal species, especially in northern African countries, despite evidence that they can be a potential source of human contamination. The study reported here compared two different methods to assess the presence of antibodies against *T. gondii* in a cohort of hunting dogs from northern Algeria, as these dogs are in close contact with wildlife and humans, in both rural and urban environments. The results of this study show that about 37% of hunting dogs are infected by *T. gondii* and that both tests can be used for this diagnostic purpose. This work also highlights the importance of this kind of study in companion animals to assess infectious risk of *T. gondii* for human populations.

**Abstract:**

Toxoplasmosis is a zoonotic disease, caused by the protozoan *Toxoplasma gondii*, affecting most warm-blooded animals. Assessing the seroprevalence of *T. gondii* in different animal species gives a good estimate of the global circulation of the parasite and the risk for human infections. However, the seroprevalence of *T. gondii* in dogs is not studied as much as other species, despite their close contact with wildlife and humans in rural or urban environments and evidence that dogs can also be a potential source for human contaminations. A commercial enzyme-inked immunosorbent assay (ELISA) kit to detect anti-*T. gondii* antibodies in sera of hunting dogs potentially naturally infected, was compared to the modified agglutination test (MAT), used as the reference method. The ELISA presented a sensitivity of 76.5% (CI 95%: 60.0–87.6) and a specificity of 87.7% (CI 95%: 76.7–93.9) and a substantial agreement with the MAT for the detection of canine anti-*T. gondii* antibodies. Both tests can therefore be used widely for epidemiology studies on *T. gondii* infections in dogs. With a mean seroprevalence of *T. gondii* infection in hunting dogs from northern Algeria of 36.8% (CI 95%: 34.9–38.7), this study also highlights the importance of *T. gondii* seroprevalence studies in companion animals to assess infectious risk for human populations.

## 1. Introduction

Toxoplasmosis is among the many parasitic diseases that are a challenge for public health. This zoonosis affects most warm-blooded animals, including birds, mammals, and humans [1,2,3]. It is caused by a protozoan of the Apicomplexa branch, *Toxoplasma gondii* (*T. gondii*), and is of economic, veterinary, and medical importance. The asexual cycle of *T. gondii* can occur within any warm-blooded animal (intermediate hosts), but its sexual cycle develops only in felines (definitive hosts). Intermediate hosts can be infected in different ways, either by consumption of undercooked or raw meat containing tissue cysts, or by food and water contaminated with oocysts shed by the definitive host, or by transplacental infection by tachyzoites [4].

In most cases, infection in humans is asymptomatic or subclinical, but severe forms can occur in immunocompromised individuals and in the fetuses or the newborn, as a consequence of infection during pregnancy [5]; therefore, prevention of toxoplasmosis is of public health importance. To achieve this goal, it is important to know the serological status of humans for *T. gondii* infection, but also in production animals for risk assessment of potential contamination [6]. Serological status of companion animals is also important to assess as cats are the definitive host of the parasites. In dogs, *T. gondii* infectious oocysts have also been identified in feces after either experimental [7] or natural [8] infections, indicating a passage of the oocysts through the dog gut without degradation, suggesting the risk of re-shedding in a different environment. They can also carry oocysts on their fur, which can then be transmitted to humans [9,10,11]. Therefore, in addition to cats and contaminated food, dogs represent a rare but real potential source of infection to humans. Moreover, dogs, and mainly hunting dogs, represent good sentinels for environmental and wildlife contaminations because they live in close contact with both humans and wildlife in rural or urban environments.

The serodiagnostic tests most commonly used both in clinical and epidemiological surveys for the screening of infection of humans and animals with *T. gondii*, include the enzyme-linked immunosorbent assay (ELISA) and the modified agglutination test (MAT) [4,12]. Both tests have shown good sensitivity and specificity for detecting a chronic *T. gondii* infection (IgG), in humans and animals but often in veterinary studies, MAT is preferred, as it is simple, cost-effective, and there is no species limit as compared to ELISA; however, it can produce non-negligeable false negative results in dogs [13]. Therefore, we aimed in this study to detect *T. gondii*-specific antibodies in naturally exposed hunting dogs from different regions of Algeria by using a commercial ELISA kit, and compare results with those obtained with the MAT. The latter test was used here as the reference standard, previously developed and used for epidemiological studies on *T. gondii* seroprevalence in many different species by us and others [4,14,15,16].

## 2. Materials and Methods

### 2.1. Sampling 

Ninety-one blood samples were collected from hunting dogs in three districts of the north of Algeria (Blida, Boumerdes, Algiers) between February and September 2019. The study cohort was composed of 40 males and 51 females, and out of the 91 dogs, 53 were under two years old. Fifty-three dogs were sampled in Algiers region, 19 from the Blida region and 19 from the Boumerdes region. For each animal, a volume of 3 mL blood was collected from the forelimb vein (scalp vein set 21G × 3/4, Nipro Europe N.V. Zaventem, Belgium) into a dry tube, and centrifuged at 3000 rpm for 10 min after clotting. The separated sera were stored in cryotubes at −20 °C until further analysis. 

### 2.2. Ethical Statement 

The study was approved by the scientific and pedagogical council of the Higher National Veterinary School, Algiers, Algeria, in accordance with the Algerian legislation (Ordinance N° 06–05 of 19 Joumda Ethania 1427 corresponding to 15 July 2006). The sampling was performed under the laws N° 88–08 of the 26th of January 1988 and N° 19–03 of 14 Dhou El Kaâda 1440 corresponding to the 17 July 2019, concerning activities of veterinary medicine and animal health protection. Dogs belonging to hunters were sampled by the veterinarian with the consent of the owners who were informed about the purpose of the study and gave their agreement to participate in this investigation. 

### 2.3. Modified Agglutination Test (MAT)

The MAT was performed as previously described [17]. Briefly, two-fold dilutions (1:6 to 1:48) of the sera and controls were made with PBS 1X (Gibco, Thermofisher, Illkirch, France) in a 96-well microtiter plate (Greiner Bio-One GmbH, Kremsmünster, Austria). Negative and positive canine control samples were obtained from previously agreed experimental infections, with the positive serum presenting a titer of 1:1536. Then, 3 × 10^5^ *T. gondii* particulate antigens (kindly provided by National Reference Center for Human Toxoplasmosis, Université de Reims, Champagne, Ardenne, Reims, France) diluted in bovine albumin buffer solution (BioMérieux, Craponne, France) were distributed in each well with 6.6 mM dithiotreitol (DTT, Sigma Adrich, Saint-Quentin-Fallavier, France). Agglutination of *T. gondii* antigens was then observed after incubation at room temperature for 16 h to 18 h. Positive sera were further analysed at higher dilutions (1:6 to 1:12,288) to establish the endpoint titer. The antibody titer was determined by the last dilution of the sample, allowing the positive reaction to be observed: agglutination covering at least 50% of the surface of the well. The positivity threshold was set at the titer of 1:6, as previously published [17]. In order to avoid potential bias, results of the MAT were blind analysed by two different assessors. 

### 2.4. Enzyme-Linked Immunosorbent Assay (ELISA)

Serum anti-*T. gondii* antibodies were measured with a specific commercial ELISA kit designed for multiple species (ID Screen Toxoplasmosis Indirect Multi-species, IDVet, Grabels, France) following the manufacturer’s instructions. Briefly, pre-coated 96-well plates with P30 antigen from *T. gondii* were incubated for 45 min at room temperature with dog sera and standards, all diluted at 1/10 in sample buffer. After washes and revelation steps, plates were read at 450 nm, on the MultiskanFC reader and analysed with SkanIt Research Edition 4.1 software (ThermoFisher Scientific, Illkirch, France), according to manufacturer’s instructions. For each sample, the S/P percentage (S/P%) was calculated according to the following formula given in the kit: S/P% = (OD_sample_/OD_pc_) × 100, where S means serum sample, P means positive sample, OD_sample_ is the optical density of sample, and OD_pc_ is the optical density of positive control. The interpretation was as follows, according to the manufacturer instructions for canine sera: S/P% ≥ 70% is a positive result, 40% < S/P% < 70% is a doubtful result, and S/P% ≤ 40% is a negative result. 

### 2.5. Statistical Analysis 

The seroprevalence was calculated from the ratio of the positive samples to the total number of sera tested, with the exact binomial confidence interval of 95% (CI 95%). The positive and negative results obtained by MAT and ELISA were also classified into two-by-two contingency tables, with MAT considered as the standard. The Fisher’s exact test was used to analyze the difference in efficiency, and *p* < 0.05 was considered significant. The agreement between MAT and ELISA to detect *T. gondii*-specific antibodies was evaluated by using the Cohen’s kappa coefficient (k). The strength of the agreement between the two techniques was assessed as follows: <0, no agreement; 0–0.2, small agreement; 0.2–0.4, fair agreement; 0.4–0.6, moderate agreement; 0.6–0.8, substantial agreement; 0.8–1, almost perfect agreement [18]. Relative sensitivity, specificity, positive predictive value and negative predictive values of the ELISA test with respect to MAT as the standard, were calculated as corresponding descriptive test parameters. All analyses were performed with GraphPad Prism (version 9.3.0 (463); Dotmatics, Boston, MA, USA).

## 3. Results

Hunting dogs assessed for *T. gondii* antibodies were sampled from three regions in the vicinity of Algiers, Algeria, representing peri-urban and rural areas of Northern Algeria (Figure 1).

Among 91 tested sera by MAT, specific IgG anti-*T. gondii* were found in 34 samples (37.4%; CI 95%: 34.3–40.4). Forty-seven percent (47%; CI 95%: 45.6–48.4) of female dogs were seropositive by MAT when only 25% (CI 95%: 23.4–26.6) of males were seropositive. Regardless of sex, *T. gondii* seroprevalence was 30.2% (CI 95%: 28.6–31.4) in dogs sampled in Algiers (region 1) and a similar seroprevalence (36.8%; CI 95%: 33.8–38.3) was found in dogs sampled in Boumerdes (region 3). However, a higher seroprevalence was detected in dogs from Blida (region 2) with 57.9% (CI 95%: 54.8–59.3) of dogs seropositive by MAT. By using ELISA, 33 dog samples (36.2%; CI 95%: 33.2–39.3), were found to be positive, giving an overall seroprevalence of 36.8% (CI 95%: 34.9–38.7) in hunting dogs from the three regions sampled. By ELISA, seroprevalence in female dogs was close to seroprevalence of male dogs with respectively 35.3% (CI 95%: 35.2–37.3) and 37.5% (CI 95%: 35.9–39.0). *T. gondii* seroprevalence was 24.5% (CI 95%: 20.7–28.2) in dogs sampled in Algiers (region 1). However, a slightly higher seroprevalence was detected in dogs from Boumerdes (region 3, 42.1%; CI 95%: 41.4–42.8) compared to MAT results. As with MAT, dogs from Blida (region 2) exhibited a higher seroprevalence of 63.1% by ELISA (CI 95%: 62.0–64.0). The results obtained for all samples with both techniques are presented in Figure 2 and contingency Table 1. 

Twenty-six sera tested positive by both techniques. A discrepancy was found for seven samples which were positive for ELISA but negative in MAT, and eight samples were negative for ELISA while positive with MAT. 

We therefore performed a comparison of the two techniques focusing on positive samples in both ELISA and MAT to evaluate which MAT titers most corresponded to a positive signal in ELISA. Table 2 shows positive samples with a MAT titer between 1:6 and 1:768. Of the 26 positive samples, nine ELISA positive sera are positive with MAT at a titer of 1:12, and six ELISA positive sera had a MAT titer of 1:24. However, it is important to notice that six sera had a high titer, above 1:96 with one reaching the maximum titer of 1:768. 

Table 2 shows also MAT positive titers of the eight samples, which were negative by ELISA (S/P% ≤ 40%, as set by manufacturer). Most of these samples present a low MAT titer (62.5%); however, two of them have a non-negligible MAT positive titer. This table does not show the seven samples which were ELISA (+) but MAT (−). Among these, all seven samples exhibited a S/P% value above the threshold of 70% set by the manufacturer, and therefore were not considered as doubtful by ELISA. 

The Fisher’s exact test was then used to assess the sensitivity and specificity of ELISA which were of 76.5% (CI 95%: 60.0–87.6) and 87.7% (CI 95%: 76.7–93.9), respectively, with MAT as standard. A substantial agreement (k = 0.646) was observed between the two methods. All of the comparative test parameters of ELISA with MAT are presented in Table 3. 

## 4. Discussion

In this study, we determine the seroprevalence of *T. gondii* infection in a cohort of hunting dogs in Northern Algeria, and for this we used a commercial ELISA, the results of which were compared to the reference method MAT. 

Canine *T. gondii* infection is not often assessed in Algeria and in other northern African countries, as reported [4,19,20]. The specific population of hunting dogs is particularly exposed to *T. gondii* infections through consumption of raw meat and exposure to the pathogen in the environment (by ingestion of oocysts in contaminated soil, water, or cat feces directly). Therefore, we focused on this specific dog population, which are more likely to be naturally infected or exposed to the parasite, as compared to companion dogs, as they are close to wildlife and environment but also to a dense urban human population in these specific three regions of Algeria. The results showed that the seroprevalence of *T. gondii* in hunting dogs was 37.4% (CI 95%: 34.3–40.4) with MAT and 36.2% (CI 95%: 33.2–39.3) with ELISA, giving a mean value of 36.8% (CI 95%: 34.9–38.7). The only previous study on *T. gondii* seroprevalence in 105 dogs from Algeria, using the complement-fixation technique, had identified a value of 30.5% [20,21]. *T. gondii* seroprevalence in this hunting dog cohort is close to this former value from 1955, but comparison of these values is hard to assess because sensitivity and specificity of the techniques are different, as well as the study population. However, results from the present study may suggest that dogs, when in contact with wild animals or the environment, can be infected with *T. gondii*, therefore supporting the assessment that *T. gondii* seroprevalence in dogs is a good indicator for environmental, and possibly wildlife, contaminations by this parasite [22]. This also confirms the transmission risk of the disease that they may represent as previously described [7,8,11], despite the fact that it can be considered as a rare event as opposed to direct infection of humans by contaminated meat or vegetables. 

Serological tests are useful for epidemiological surveys, such as the ones we performed in this study. They provide evidence for actions for the control and prevention of possible contaminations of *T. gondii* in humans and animals, and they also give an estimation of the rate of infection by the parasite in different populations. The ELISA is easy to perform, and allows analysis of a large number of samples rapidly [23]. However, MAT is still widely used for *T. gondii* seroprevalence studies because it is simple, cost-effective, accurate, and can be used for a wide variety of species, as it does not require species-specific conjugates [12]. Both tests used in this study allow us to monitor *T. gondii* IgG but not IgM, which can indicate a recent and acute infection [4,24]. However, a high MAT titer associated with a high S/P% (≥200) could be a sign of an acute infection as indicated by the ELISA manufacturer and which has been seen for three of our 91 samples.

In a similar study carried out in China in companion animals [12], a significant difference between these two techniques for detecting *T. gondii* antibodies in dogs was identified, with a low prevalence detected by the MAT test (23.1%) and a prevalence judged higher with ELISA (34.7%). This could be due to the fact that MAT can sometimes give false negative results when used on canine sera [25]. However, in the present study, seroprevalences obtained with both techniques were equivalent. 

The choice of diagnostic test depends on several parameters, including, among others, their sensitivity and specificity. In this study, MAT was considered as the reference test for determining the seroprevalence of toxoplasmosis because this approach was used in epidemiological studies to validate other serological tests [23,26,27] and also because performance values of the ELISA kit were previously calculated for dog sera tested in comparison with the indirect fluorescent antibody technique (IFAT) but not with MAT. A concordance has already been reported between MAT and the detection of viable *T. gondii* in pigs [28], horses [29], and sheep [17] by bioassays reflecting the good sensitivity and specificity of the technique. In the present study, the relative sensitivity and specificity of the ELISA in comparison with the MAT, are of 76.5% (CI 95%: 60.0–87.6) and 87.7% (CI 95%: 76.7–93.9), respectively. These results were obtained considering that the results were positive with MAT, from a titer of 1:6, according to the protocol previously published [14,15,16]. These values are lower than the ones published in a study carried out on stray dogs in Brazil, with a positivity threshold higher than our study, fixed at MAT cut-off of 1:25 (with a sensitivity of 78% and a specificity of 99%) [30]. The variability in sensitivity and specificity could be due to the assay used in the different studies but may also depend on immune status, biological factors, and the susceptibility of each animal species for *T. gondii* [12]. Therefore, we aimed to evaluate the agreement between MAT and ELISA for the detection of *T. gondii* antibodies in the sera of the cohort of hunting dogs by using the Cohen’s kappa coefficient (k). The present study showed a substantial agreement between ELISA and MAT for detecting *T. gondii* antibodies in canine sera, allowing the two techniques to be suitable for epidemiological surveys in dogs. 

## 5. Conclusions

Both ELISA and MAT can be used to determine *T. gondii* seroprevalence in dogs. By using these two serological tests, the seroprevalence for *T. gondii* in a cohort of Algerian hunting dogs was of almost 37%, all localized around Algiers, indicating that the parasite still circulates in this environment. Future studies that adapt different designs allowing additional comparisons between ELISA, MAT, and other serological assays may provide further insights into the best approach for large-scale epidemiological studies on *T. gondii* infections in dogs from North Africa. Nonetheless, this study supports the need to perform more detailed studies in hunting dogs, with bigger cohorts, to investigate their route of infection and if they may excrete, or carry, infectious oocysts which could be subsequently transmitted to humans. 

## Figures and Tables

**Figure 1 animals-12-02813-f001:**
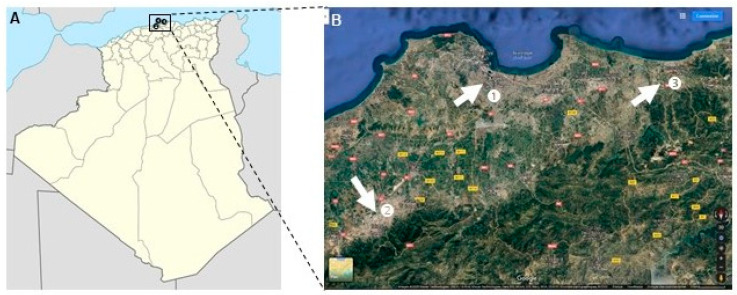
Regions of sampling for the study. (**A**) Blood of hunting dogs were collected in Algiers (1; 273 km^2^, population: 2,947,461), Blida (2; 1696 km^2^, population: 1,009,892) and Boumerdes (3; 1591 km^2^, population: 795,019) regions (black dots). (**B**) Regions 2 and 3 are surrounded by mountain landscapes and region 1 is suburban with forests and fields (white arrows, Blida is southwest of Algiers, and Boumerdes is east of Algiers, Satellite image provided by GoogleMap, Map data: Google, Maxar Technologies, as indicated).

**Figure 2 animals-12-02813-f002:**
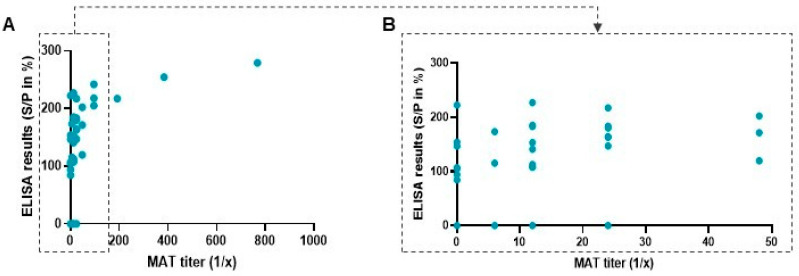
ELISA and MAT results for all dog sera tested. (**A**) Results of the 91 tested samples. (**B**) Zoom on the low MAT titer samples (between 1/6 and 1/48).

**Table 1 animals-12-02813-t001:** Contingency table for detection of *T. gondii* antibodies in sera of hunting dogs by commercial ELISA in comparison with MAT (n = 91).

	MAT (+)	MAT (−)	Total (Rate)
ELISA (+)	26	7	33 (36.2%)
ELISA (−)	8	50	58 (63.7%)
Total (rate)	34 (37.4%)	57 (62.6%)	91

**Table 2 animals-12-02813-t002:** MAT titers of ELISA positive dog sera (n = 26) and ELISA negative but MAT positive (n = 8).

	MAT (Titer)							
	1:6	1:12	1:24	1:48	1:96	1:192	1:384	1:768
ELISA (+)	2	9	6	3	3	1	1	1
(%)	7.7	34.62	23.07	11.54	11.54	3.85	3.85	3.85
ELISA (−)	5	2	1	0	0	0	0	0
(%)	62.5	25	12.5	0	0	0	0	0

**Table 3 animals-12-02813-t003:** Comparison of the ELISA to MAT to detect *Toxoplasma gondii* antibodies in a cohort of 91 serum of hunting dogs (descriptive test parameters and measures of agreement).

	ELISA	CI 95%
Sensitivity (%)	76.5	60–87.6
Specificity (%)	87.7	76.7–93.9
Positive predictive value	78.8	62.2–89.32
Negative predictive value	86.2	75.1–92.8
Likelihood ratio	6.23	
Kappa coefficient (k)	0.646	0.483–0.809

## Data Availability

Not applicable.

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
