# Peer review of "Comparison of a Commercial Enzyme-Linked Immunosorbent Assay (ELISA) with the Modified Agglutination Test (MAT) for the Detection of Antibodies against Toxoplasma gondii in a Cohort of Hunting Dogs"

_animals, 2022, doi:10.3390/ani12202813_

Round 1

Reviewer 1 Report

The manuscript deals with the comparison of two serological methods for detecting antibodies to Toxoplasma gondii in hunting dogs and provides the seroprevalence of toxoplasmosis in hunting dogs in three regions of Algeria. Although similar studies from other areas already exist in literature, the results contribute to the awareness of circulation of this important zoonotic pathogen.

General comments:

An impersonal style should be preferred, avoiding words like "our" or "we". Several typos are present throughout the text that needs to be checked.

Specific comments:

Line 31: Enzime Linked Immunosorbent Assay (ELISA) – for first mentioning

Line 70: “We have previously developed…”  “We” should not be used in this context, although some of the authors of the manuscript contributed to cited investigations - references 13, 14, 15.  

Line 98: It should be explained which testing plate was used (microtiter plate?)?

Line 99: The sentence "...eight two-fold dilutions (1:6 to 1:48)" should be corrected.

Line 189: “et al., (2019)” should be omitted.

Line 200, 215: “…our dog”, “…our seroprevalence value” My suggestion to avoid word "our" where possible.     

Line 224: For a study carried out in China reference 12 should be cited.

Lines 243-245: References for this statement should be provided.

Line 246: "substantial" agreement instead of "good" agreement

Reviewer 2 Report

Dear Authors,

The aim of this study was the evaluation of a commercial ELISA kit for the serological diagnosis of toxoplasmosis in hunting dogs, using the MAT as a gold standard.

Despite MAT is widely used for the serological diagnosis of Toxoplasmosis in many species, including humans, and some papers were published comparing MAT to ELISA, in my opinion the evaluation/validation of ELISA using MAT as a gold standard could be penalizing for ELISA. I think it would be better to redesign the study, comparing the two tests in absence of a gold standard by means of a Bayesian analysis.

The results here described are not encouraging. A K value of 0.646 cannot be considered very satisfying in the routine diagnosis, as well as the calculated Se and Sp values (respectively, 76.5% and 87.7%), especially if the serological diagnosis would be used to classify the serological status of single subjects.

The Authors evaluated in detail the 8 samples ELISA-negative and MAT-positive, showing mainly low MAT titers. What about the 7 samples that resulted ELISA-positive and MAT-negative? Did they have low ELISA S/P values, close to the cut-off? Did the Authors try to perform ROC analysis to find another potential cut-off for the ELISA test?

In detail:

Line 57-58: to assess the serological status of single subject as companion animals, a test with high values of sensitivity and specificity should be used. This point should be considered in the discussion.

Line 62-64: I agree that dogs do represent good sentinels for environmental contamination; on the other hand, dogs as a real potential source of infection for humans should be considered as a very low risk on an epidemiological point of view. Probably, eating raw vegetables is much more dangerous. I think that this point should be considered in the discussion and conclusions (line 254).

Line 80-81: why was the sampling designed in three different districts? Is it possible that in the three areas there are different prevalences? Did the Author consider this point?

Line 216: “most of positive samples in ELISA were also positive with the MAT technique”: on a practical diagnostic and statistical point of view, I do not agree with this statement

Line 246: Sorry, I would not state that the study showed a good agreement between ELISA and MAT 

Reviewer 3 Report

As mentioned by the authors few studies have been done on Toxoplasma gondii seroprevalence in dogs. This study performed on hunting dogs in Algeria is therefore of interest. Apart from some minor corrections to be made that I list below I have two general comments:

1. There is no mention in the manuscript on the Immunoglobulin class(es) detected by the two tests. Is 2-mercapto-ethanol used in the MAT (to destroy IgM that are often not specific); and is an anti-IgG conjugate used in the ELISA? In the manuscript, including the discussion, nothing is written on immunoglobulin classes and their importance in assessing acute or chronic infections.

2. The sample size is 91. This seems rather small. Is this number sufficient for  a correct estimate of the seroprevalence? Has any type of sample size calculation been used?

Other corrections:

- On many places in the manuscript MAT is followed by "test" or "technique" or "assay" while the "T" in MAT stands for "Test". Adding "test" or "assay" is redundant.

- On many places in the manuscript ELISA is followed by "assay" or "test" while the "A" in ELISA stands for "Assay". Adding "assay" or "test" is redundant.

- "wild life" should be written in one word "wildlife"

- Fig 1 could be made more clear. E.g. by adding the numbers of the localities in map B

- line 173 and on other places: MAT is not a gold standard test (defined as having 100% sensitivity and 100% specificity), it is rather the reference test in this study

- Line 253 in the Conclusions section: replace "secrete" by "excrete"

Reviewer 4 Report

General comment:

The MS entitled „Evaluation of a commercial Enzyme Linked ImmunoSorbent Assay (ELISA) for detecting antibodies against Toxoplasma gondii in a cohort of naturally infected hunting dogs” describes anti-T. gondii antibodies in hunting dogs in Algieria. Even though results may be valuable, the MS needs some improvements, explanation etc. I do not understand why Authors used in-home MAT as a standard. How was it standardized? How did they calculate specificity and sensitivity for MAT? In my opinion MAT is less accurate than ELISA, due to eye reading. Did Authors take into account false positive/negative results obtained by MAT? I recommend to reverse the way of thinking: use ELISA as a standard, and evaluate in-home MAT. I do not agree with statistic calculations – in my opinion Authors should calculate it one more time or explain widely how they calculate chi2 test.

Specific comments:

Title: Title should be changed – because it suggests that MS describes ELISA only, however the Authors showed also results obtained by in-house MAT.

Line 32: add Modified before Agglutination Test.

Lines 62-63:  Why did Authors state that dogs represent a real potential source? They did not examine it. Explain, give reference or change.

Line 75: Why Authors use MAT as a standard?

Lines 108-109: Published where? Give reference.

Lines 114-116: Does IDVet recommend to incubate their plates for 1 hour in 37 Celsius degree? In my manual is stated: “incubated at room temperature for 45 minutes”. Please explain, it may change results Authors obtained. Whether the authors only tested one replicate of each sample for both tests?

Figure 1: Authors should mark localizations on the map with: 1, 2 and 3.

Table 1. How do Authors explain these differences in ELISA and MAT? Why sometimes is ELISA positive and MAT negative, and vice versa?

Lines 171-174: Why did Authors state that MAT is a GOLD standard? Any reference?

Line 177: What specificity and sensitivity do Manufacturer give in the manual of ELISA?

Lines 203-205: “However, our results confirm that dogs, when in contact with wild animals or environment, can be infected with T. gondii, therefore making T. gondii seroprevalence in dogs, a good indicator for environmental and wild life contaminations” – on what basic did Authors state that? Did they analyze it? In my opinion their study does not confirm that dogs are a good indicator for environmental and wild life contaminations.

Lines 245-247: Authors stated that there is a good agreement between two methods. How it is possible that Authors state before that there are statistically significant differences between ELISA and MAT? It has been showed 33 positive ELISA results, and MAT only one more? Please explain.

Round 2

Reviewer 2 Report

Dear Authors,

I suggested to redesign the study under a statistical point of view, starting from the laboratory data and not from the sampling. So, I am a little bit surprised about the Author's reply to my first comment.

About the point: "The results here described are not encouraging. A K value of 0.646 cannot be considered very satisfying in the routine diagnosis, as well as the calculated Se and Sp values (respectively, 76.5% and 87.7%), especially if the serological diagnosis would be used to classify the serological status of single subjects." I share the Author's point of view, but I would like to stress that these Se and Sp values appear not satisfying to test and classify single subjects. Maybe the performance of the ELISA test could be underevaluated in this study.

About the comment; "What about the 7 samples that resulted ELISA-positive and MAT-negative? Did they have low ELISA S/P values, close to the cut-off?", maybe I did not express myself clearly, as I was asking about the S/P values of the 7 samples ELISA-pos and MAT-negative.

Furthermore, In reply to R.4 the Authors replied that "None of the other 3 reviewers suggested to completely modify the design of the study in this way". It is true, but the the study design maybe should be reconsidered in order to better evaluate the ELISA performances.

Reviewer 4 Report

Line 30: I do not agree that T. gondii seroprevalence is rarely studied in dogs. It has been studied eg. in Argentina, Brasil, China, Iran, Japan, Mexico, Nigeria, Panama, Poland, Spain, Thailand, Turkey, USA. Why Authors did not discuss these results in Discussion section? Moreover, it would be valuable to discuss results concerning on hunting dogs eg. from Brasil, Italy, Taiwan etc. Additionally, seroprevalence of T. gondii antibodies in dogs has been studied: Angola, Egypt, Gabon, Madagascar, Nigeria, Senegal. Why the Authors did not discuss it?

Line 36: Do you mean: substantial?

Lines 112, 162-165, 226: values as 6,6, 30,2 should be written as 6.6, 30.2, etc.

Lines 143-144: Add agreement after: small, moderate etc.

Lines 162-168: “Regardless of sex, T. gondii seroprevalence was 30,2% (CI95% : 28.6 – 31.4) in dogs sampled in Alger (region 1) and a similar seroprevalence (36,8% ; (CI95% : 33.8 – 38.3) was found in dogs sampled in Boumerdes (region 3). However a higher seroprevalence was detected in dogs from Blida (region 2) with 57,9% (CI95% : 54.8 - 59.3) of dogs seropositives by MAT.”

The Authors derived results from three regions obtained only by MAT. As the Authors compare two assays, it should be done also for ELISA.

Line 169: Capital letter dot Figure 2 and Table 1

Line 192: Do you mean: as doubtful?

Lines 229-230: “However, our results from the present study confirm that dogs, when in contact with wild animals or environment, can be infected with T. gondii,”

Are the Authors sure that hunting dogs were infected because they were “in contact with wild animals or environment”? I suggest: “However, results from the present study may suggest that dogs, when in contact with wild animals or environment, can be infected with T. gondii…”

Lines 248-251: this study also suggest “serum-based ELISA would be more satisfactory for epidemiological survey of toxoplasmosis in dogs”. The Authors did not discuss this conclusion.

Lines 264-265: But it was eg. in wolverines. It would be valuable to discuss it.
